# Impact of Natural Forest Succession on Changes in Soil Organic Carbon in the Polish Carpathian Mountains

**Justyna Sokołowska, Agnieszka Józefowska \*** and **Tomasz Zaleski**

Department of Soil Science and Agrophysics, University of Agriculture in Krakow, al. Mickiewicza 21, 31-120 Kraków, Poland; justyna.sokolowska@student.urk.edu.pl (J.S.); tomasz.zaleski@urk.edu.pl (T.Z.)
\* Correspondence: agnieszka.jozefowska@urk.edu.pl

**Abstract:** The main driver of the Carpathian landscape is the process of natural forest succession on the semi-natural meadows unique to the region. Moreover, these semi-natural mountain meadows contribute to ecosystem services, although increasing forest areas are recommended by current international policy agendas. The purpose of this study was to examine the impact of natural forest succession in the Polish part of Carpathian on changes in soil organic carbon and assess the influence of different soil properties on organic carbon content across three land uses. Soil samples were taken from 10 transects consisting of semi-natural mountain meadow, natural successional forest and old-growth forest, selected in three Polish Carpathian national parks. Measurements of organic carbon, dissolved organic carbon, microbial properties, such as microbial respiration, and enzyme activities were made; additionally, biochemical indicators were calculated. To describe the influence of measured soil parameters and calculated indicators of soil organic carbon changes, the organic carbon dependent variable regression equations across all studied soils and for the individual land use and examined layers were evaluated. The overall regression equation indicated that changes in organic carbon general to all investigated soils depended on microbial biomass carbon content, microbial quotient, dissolved organic carbon content and metabolic quotient. The regression models obtained for the individual land use variants and soil layers explained 77% to 99% of the variation in organic carbon. Results showed that natural forest succession caused a decrease in microbial biomass carbon content, and successional forest soils characterized less efficient use of organic substrates by microbial biomass.

**Keywords:** ecosystem services; landscape transformation; protected areas management; microbial activity; microbial respiration

## 1. Introduction

European mountain areas are characterized by the high diversity of habitats resulting from natural conditioning as well as human activity. The Carpathian Mountains are the largest, longest and most fragmented mountain chain in Europe. Moreover, this mountain chain is characterized by a high variety of relief, with intra-mountainous regions, depressions, sub-mountain hills and lowlands. These, along with altitudes exceeding 2500 m, contribute to the magnificent landscape of the Carpathians [1].

Semi-natural mountain meadows are an integral part of the Carpathian landscape [2]. In recent times, however, less-productive and inconveniently located grasslands are being increasingly abandoned [3] due to socio-economic changes in the Carpathians. Following abandonment, overgrowing of these unique meadows by the surrounding forest has begun to progress. The negative and positive effects of these land-use changes and continuous natural forest succession in the Polish part of Carpathians are a debatable issue, especially in the context of carbon sequestration and climate changes. In any case, semi-natural mountain meadows have a great contribution to ecosystem services. These areas provide food, e.g., hay for sheep; regulating services, e.g., water regulation, preventing soil erosion

and carbon sequestration; supporting services, such as primary production and water cycling; and cultural services, e.g., tourism and contributing to the overall landscape. Moreover, it is well known that land-cover changes influence carbon cycling in the soil. Thus, the Polish part of the Carpathian Mountains is an appropriate study area to assess changes in the soil carbon cycle during natural forest succession.

The importance of such studies was highlighted by Garcia-Pausas et al. [3], who stated that woody plants can significantly influence carbon balance, but the size of this effect is still unknown. Moreover, soils contain more than twice the carbon found in the atmosphere [4]; thus, even a little change in soils may greatly influence the carbon balance. Many authors [5–7] have reported the impact of land-use changes on organic carbon content in soil. Nevertheless, land-use changes simultaneously may cause carbon sequestration or carbon dioxide emission. Restoring grasslands, forest or other native vegetation on former croplands increases soil organic carbon content [4]. However, there is still a lack of knowledge covering the impact of natural processes, such as forest succession on soil organic carbon.

Some of the main factors that influence soil organic carbon changes are biotic properties, consisting mainly of the quantity and quality of carbon inputs into soil. These factors work together with climatic variables and abiotic soil factors to regulate carbon dynamics in soils [8]. Natural forest succession processes in the Carpathian make it possible to distinguish different kinds of carbon inputs, such as easily decomposed grass in meadows, fresh fall from young, successional trees and the constant supply of fresh litter during the climax stage of a forest. As such, studies on soil organic carbon changes in a changing mountain environment may be particularly useful.

Soil is a dynamic environment in which accumulation and decomposition of organic matter continuously occur. Two crucial processes leading to transformation of soil organic matter are mineralization and humification. According to Kandeler et al. [9], microorganisms play an important role in organic matter decomposition, especially mineralization, mainly via respiratory processes. Measurement of soil heterotrophic respiration, referring to the production of $CO_2$ from microbial respiration [10], is simply a method to describe overall soil conditions, as well as the activity of the microbial community in soil [11]. This process has an essential contribution to carbon cycling and the global carbon budget by releasing carbon dioxide from the soil to the atmosphere [12].

Studies on soil heterotrophic and autotrophic respiration were used to predict soil carbon processes and below-ground carbon sequestration in successional forests [13]. Microbial soil respiration was also broadly used in the studies to monitor changes between different land-use areas, such as grasslands converted to cornfields [14], wheat fields, vineyards and cherry farms [15], or natural forest, arable lands, citrus gardens and paddy fields [16], but studies on natural soil-respiration changes in land-use transformation are very rare. Soil heterotrophic respiration is usually used in combination with other indicators, such as microbial biomass carbon as an indicator of soil quality. Some authors [17,18] have highlighted the importance of quantifying soil microbial activity parameters as indicators enhancing the evaluation of changes in land use.

Additionally, other parameters, such as metabolic quotient, microbial quotient and mineralization quotient, have been evaluated to monitor the changes in soil quality connected with soil organic carbon cycle [16,19,20]. The metabolic quotient was used as an index of microbial efficiency in utilizing the available resources [18], as well as an indicator of the degree of substrate limitation for soil microbes [21]. The microbial quotient reflects the microbial biomass contribution to soil organic carbon and also indicates the fraction of recalcitrant organic matter in the soil [20]. Moreover, the microbial quotient was characterized as an indicator of further changes in organic matter during land-use alterations [22]. The mineralization quotient expresses the fraction of total organic carbon mineralized during the incubation time [20] and was used as an indicator of condition stability of chemical, biochemical and microbiological properties [16], as well as being characterized as an indicator of the efficiency of micro-flora in metabolizing soil organic

carbon [23]. Nevertheless, Mganga et al. [19] defined the mineralization quotient as the most sensitive indicator of land-use change from natural ecosystem to agroecosystem.

These parameters are widely used in research as responsive indicators of soil quality, which may help to explain ecological processes of the environment [24]. Thus, in this study, metabolic quotient, microbial quotient, mineralization quotient and chosen microbial soil properties were used as the indicators to help assess and explain soil organic carbon changes during the natural forest succession in the Polish part of Carpathians. The purposes of this study were to (i) examine the impact of natural forest succession on the soil organic carbon content, (ii) assess the differences of measured soil properties and biochemical indicators in semi-natural mountain meadows, natural succession forest and old-growth forest and (iii) investigate the soil properties and biochemical parameters affecting soil organic carbon content across changing land uses.

## 2. Materials and Methods

### 2.1. Research Area

The study area was located in the Carpathian Mountains, covering the southern part of Poland. Soil samples were collected from the three Polish Carpathian national parks—Pieniny National Park (PNP), Bieszczady National Park (BdNP) and Magura National Park (MNP)—which are located in different parts of the Polish Carpathians (Figure 1).

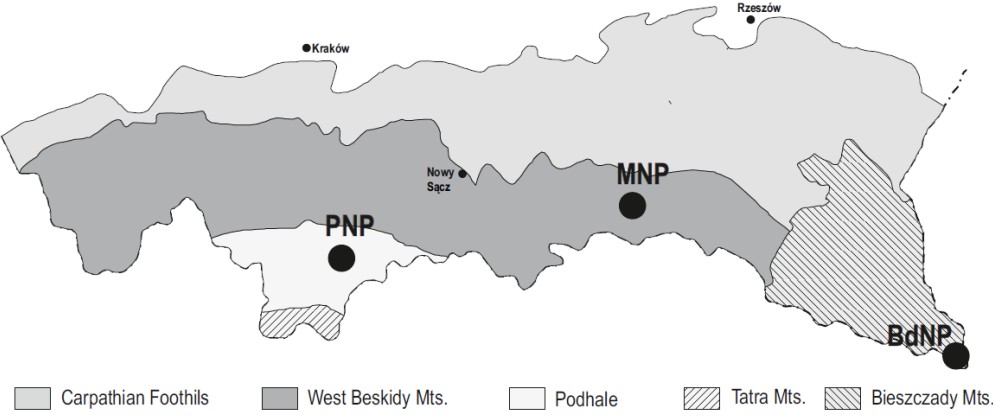

**Figure 1.** The location of the examined national parks in the Polish part of the Carpathian Mountains. BdPN—Bieszczady National Park, MNP—Magura National Park, PNP—Pieniny National Park.

The Carpathian Mountains are characterized by diversity of geological cover; in the PNP, one of the dominant parent materials are rocks of sandstone–shale series with an admixture of carbonate binder [25], whilst the BdNP and MNP soils were formed on the Carpathian Flysch with the dominance of sandstones, mudstones and shales [26]. Almost all of the investigated soils of PNP, BdNP and MNP were classifiable as cambisols with a varied trophicity (dystric and eutric cambisols) (Table 1) and were classified as loamy [27]. The basic soil properties of examined soil depended mainly on location. The highest content of total nitrogen and higher pH values were found in PNP compared to BdNP and MNP. However, BdNP characterized the highest C:N ratio [27].

**Table 1.** Location of the study areas and soil units at the area of the transects according to World Reference Base for Soil Resources (WRB) classification (IUSS Working Group WRB, 2015).

| Location | | Number of Transect | Land Use | GPS Co-Ordinates | Soil Unit (WRB) |
|---|---|---|---|---|---|
| Eastern Outer Carpathians | Bieszczady National Park | 1 | Meadow | 49°06′22.0″ N 22°43′50.5″ E | Dystric Endoskeletic Endostagnic Cambisols |
| | | | Succession | 49°06′20.1″ N 22°43′48.2″ E | Dystric Endoskeletic Cambisols |
| | | | Forest | 49°06′27.8″ N 22°43′55.0″ E | Dystric Leptic Skeletic Cambisols |
| | | 2 | Meadow | 49°07′51.3″ N 22°35′49.1″ E | Dystric Endoskeletic Amphistagic Cambisols |
| | | | Succession | 49°07′45.0″ N 22°35′37.1″ E | Dystric Endoskeletic Cambisols |
| | | | Forest | 49°08′00.9″ N 22°36′05.0″ E | Dystric Cambic Skeletic Leptosols |
| | | 3 | Meadow | 49°06′51.1″ N 22°34′12.8″ E | Dystric Skeletic Epistagnic Cambisols |
| | | | Succession | 49°06′52.3″ N 22°34′14.5″ E | Dystric Skeletic Cambisols |
| | | | Forest | 49°06′54.2″ N 22°34′34.2″ E | Dystric Endoskieletic Leptic Cambisols |
| | | 4 | Meadow | 49°03′20.4″ N 22°41′41.4″ E | Eutric Cambisols |
| | | | Succession | 49°03′17.1″ N 22°41′18.1″ E | Dystric Endoskeletic Cambisols |
| | | | Forest | 49°03′21.8″ N 22°40′35.9″ E | Dystric Skeletic Cambisols |
| Western Outer Carpathians | Magura National Park | 1 | Meadow | 49°27′53.8″ N 21°29′02.3″ E | Eutric Endoskeletic Cambisols |
| | | | Succession | 49°27′46.4″ N 21°29′04.8″ E | Eutric Skeletic Cambisols |
| | | | Forest | 49°27′33.0″ N 21°28′28.2″ E | Dystric Skeletic Cambisols |
| | | 2 | Meadow | 49°26′43.6″ N 21°29′38.2″ E | Dystric Skeletic Cambisols |
| | | | Succession | 49°26′25.3″ N 21°29′55.8″ E | Dystric Endoskeletic Cambisols |
| | | | Forest | 49°26′21.3″ N 21°29′35.5″ E | Dystric Cambisols |
| | | 3 | Meadow | 49°28′54.2″ N 21°25′23.3″ E | Eutric Skeletic Cambisols |
| | | | Succession | 49°28′48.3″ N 21°25′14.0″ E | Dystric Enodskieletic Cambisols |
| | | | Forest | 49°28′48.4″ N 21°24′46.2″ E | Eurtic Skleletic Cambisols |
| Western Inner Carpathians | Pieniny National Park | 1 | Meadow | 49°25′31.8″ N 20°25′28.0″ E | Eutric Endoskeletic Cambisols |
| | | | Succession | 49°25′35.3″ N 20°25′31.1″ E | Eutric Endoskeletic Cambisols |
| | | | Forest | 49°25′34.6″ N 20°25′33.4″ E | Eurtic Skleletic Cambisols |
| | | 2 | Meadow | 49°25′30.1″ N 20°25′03.1″ E | Eutric Endoskeletic Cambisols |
| | | | Succession | 49°25′28.5″ N 20°25′11.4″ E | Eutric Cambic Leptosols |
| | | | Forest | 49°25′30.1″ N 20°25′08.6″ E | Eutric Endoskeletic Cambisols |
| | | 3 | Meadow | 49°25′49.9″ N 20°19′36.8″ E | Eutric Cambisols |
| | | | Succession | 49°25′48.7″ N 20°19′38.4″ E | Endoeutric Cambisols |
| | | | Forest | 49°25′48.9″ N 20°19′39.5″ E | Endoeutric Cambisols |

*2.2. Sampling Scheme*

In each national park, three (PNP, MNP) or four (BdNP) transects consisting of semi-natural meadow, natural succession forest and old-growth forest were chosen (Table 1). These areas were chosen based on the historical data about land use of the study area and available satellite maps covering the study area.

The semi-natural meadows occur at sites whose natural vegetation is forest and were created as a result of forest clearing during the colonization of the Polish part of Carpathians in the 13th and 14th centuries. Over centuries, the semi-natural meadows were used for agriculture, especially pasture activity. They are formed mostly by native species of perennial plants, which have spread across the landscape because of human activity [28]. The natural succession forests are areas covered by 25–70-year-old forest and were created by overgrowing abandoned meadows. They are formed by different successional tree

species and are dominated by *Fagus sylvatica* L. (common beech). The old-growth forests are areas covered by more than 150-year-old forests. The study old-growth forest stands belong to a *Dentario glandulosae-Fagetum* (Carpathian beech forest) forest habitat.

In each of the selected areas (semi-natural meadow, succession forest, and old-growth forest) in each of 10 study transects, five soil samples from both 0–10 cm and 10–20 cm soil layers were taken. Soil samples were collected using a metal five-centimetre-diameter core.

### 2.3. Laboratory Analyses

In the laboratory, fresh soil samples were sieved with a 2 mm mesh, and a portion of each soil sample was kept at $-21$ °C prior to laboratory analysis. One week prior to the analyses, soil samples were preincubated at a temperature of 25 °C and a soil moisture value equal to 40–60% water holding capacity was achieved. The rest of each soil sample was air-dried and also sieved with a 2 mm mesh.

Microbial respiration, microbial biomass carbon (MBC), dissolved organic carbon (DOC), dehydrogenase activity (DHA) and invertase activity (INV) were assessed in the fresh soil samples. Microbial respiration was estimated by the incubation method. First, 10 g of fresh soil was placed into a 100 mL vial containing a small beaker of 3 mL 0.5 M NaOH as a carbon dioxide trap. Next, the vial with the fresh soil/NaOH beaker was hermetically closed for a three-day sampling period. Incubation took place at a constant temperature of 25 °C. After the three-day sampling period, the vial was opened and the carbon dioxide evolved from soil was quantified by titration with 0.05 M HCl after the addition of 2 mL of $BaCl_2$. After this, the vial was opened and ventilated for four days. Blank samples, i.e., vials containing only beakers with NaOH, were used to assess carbon dioxide trapped during incubation from the air closed in the vials and during handling. The microbial respiration measurements were repeated once a week for five weeks.

MBC was evaluated using a fumigation–extraction method [29]. DOC content was measured following extraction using 5 mM $CaCl_2$ (soil:$CaCl_2$ ratio 1:10) and filtered by 0.45 μm. MBC and DOC were measured using the dry combustion method with a Euro Thermo TOC-TN 1200 (Landsmeer, The Netherlands). DHA levels were assessed using the method presented by Casida et al. [30], while INV activity was determined as previously described Frankenberger and Johanson [31]. Enzyme activities were measured using a Shimadzu UV-1800 (Kyoto, Japan) spectrophotometer. In air-dried soil samples, the content of total organic carbon ($C_{org}$) according to the oxidation and reduction Tiurin method was measured [32].

### 2.4. Calculations and Statistical Analyses

Based on the first-order kinetic model of microbial respiration ($C_m = C_0(1 - e^{-kt})$), the calculated cumulative value of mineralized carbon during incubation time (35 days) ($C_m$), potentially mineralizable carbon ($C_0$) and rate constant (k) were estimated [33].

Based on the obtained results of microbial parameters and carbon content, the following biochemical indicators were calculated: (1) metabolic quotient ($qCO_2$), expressing the ratio of microbial respiration ($C$-$CO_2$) to soil microbial biomass carbon (MBC) ($qCO_2 = C$-$CO_2$/MBC) [34]; (2) the microbial quotient (qMIC), representing the quotient of soil microbial biomass carbon (MBC) and total organic carbon content ($C_{org}$) (qMIC = MBC/$C_{org}$); and (3) the mineralization quotient (qM), estimated as the ratio of cumulative respiration ($CR_{5weeks}$) and total organic carbon content ($C_{org}$) (qM = $CR_{5weeks}$/$C_{org}$) [16].

Statistical analyses were performed using Statistica 13.0 software. Means and standard errors were estimated for individual land-use variants (meadow, succession forest and old-growth forest) for both the 0–10 cm and 10–20 cm layers. To describe the influence of measured soil parameters and calculated indicators of soil organic carbon changes, the $C_{org}$ dependent variable regression equations across all studied soils and for the individual land-use and examined layers were evaluated. Moreover, differences in examined soil properties and calculated indicators for the different land-use variants and soil layers were assessed using a one-way ANOVA post hoc Tuckey's test at a significance level of $p = 0.05$.

## 3. Results

### 3.1. Total Organic Carbon and Labile Carbon

The mean $C_{org}$ content slightly increased from semi-natural meadow soils (MS) to succession forest soils (SS) to old-growth forest soils (FS) in the 0–10 cm layer, while, in the 10–20 cm layer, SS had the lowest $C_{org}$ and FS the highest. Across all land uses the content of $C_{org}$ in the 0–10 cm layer was significantly higher compared to the 10–20 cm layer. (Figure 2). In the 0–10 cm layer, the lowest DOC values were found in SS and the highest in FS, while, in the 10–20 cm layer, the mean DOC content increased from MS to SS (Figure 2). SS had the lowest MBC content in both examined layers, but only in the 10–20 cm layer was the MBC significantly higher in MS compared to SS (Figure 2).

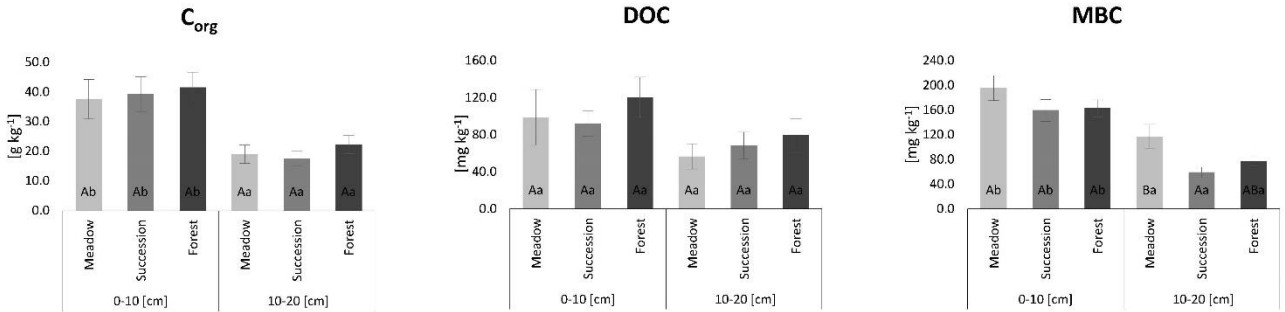

**Figure 2.** Content of total organic carbon ($C_{org}$), dissolved organic carbon (DOC) and microbial biomass carbon (MBC) in soils. The lowercase letters indicate differences between examined soil layers for individual land-use variants, while uppercase letters show differences between land-use variants separately for the 0–10 and 10–20 cm soil layers (Tukey post hoc test, $p < 0.05$). Standard error is reported as bars.

### 3.2. Microbial Activity

Enzyme activity patterns differed depending on the examined enzyme (Figure 3). DHA decreased from MS to SS to FS in the 0–10 cm layer but had the highest mean values for MS and the lowest for SS in the 10–20 cm layer. FS had the highest INV activity and SS the lowest activity in both the 0–10 and 10–20 cm layers (Figure 3).

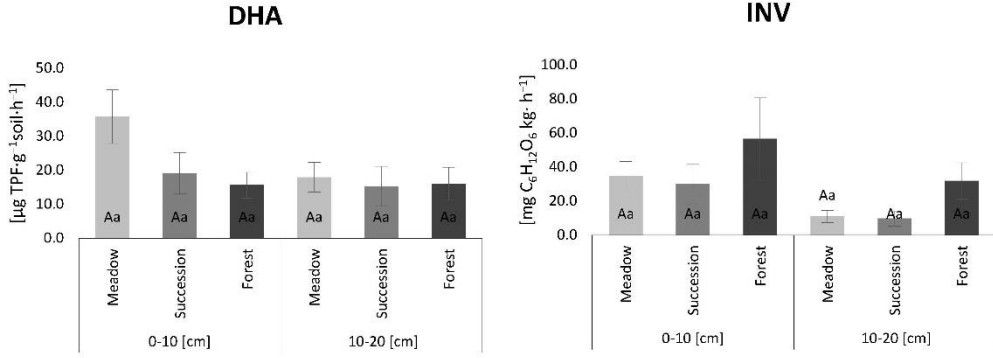

**Figure 3.** Dehydrogenase activity (DHA) and invertase activity (INV) results. The lowercase letters indicate differences between examined layers for individual land-use variants, while uppercase letTable 0 and 10–20 cm layers (Tukey post hoc test, $p < 0.05$). Standard error is reported as bars.

The mean microbial respiration ranged from 0.09 mg $CO_2$ g soil$^{-1}$ 24 h$^{-1}$ (FS, 0–10 cm, 5th week) to 1.57 mg $CO_2$ g soil$^{-1}$ 24 h$^{-1}$ (MS, 0–10 cm, 1st week). Overall, the microbial respiration values decreased with the incubation period. In the 0–10 cm layer, SS had the highest microbial respiration values in almost all incubation weeks, except the first week. In the 10–20 cm layer, over the first three weeks, the highest microbial respiration values

were found in SS; however, in the fourth and fifth weeks, the highest values were noted in MS (Figure 4).

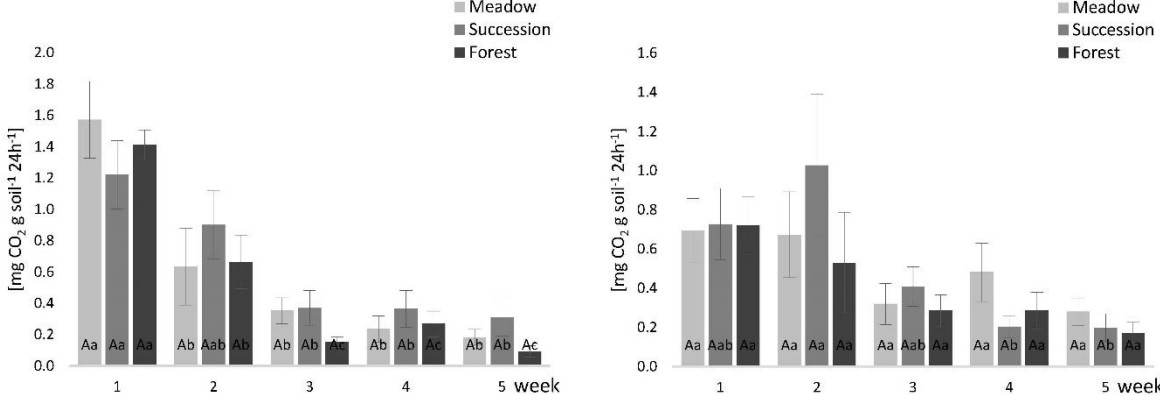

**Figure 4.** Soil respiration results for the (**a**) 0–10 cm and the (**b**) 10–20 cm soil layers. The lowercase letters indicate differences between individual weeks in the same land-use variants, while uppercase letters show differences between land-use variants for individual weeks (Tukey post hoc test, $p < 0.05$). Standard error is reported as bars.

The calculated values of mineralized carbon during the incubation time ($C_m$) and potentially mineralizable carbon pools ($C_0$) did not differ significantly between individual land-use variants as layers (Table 2). However, in the 0–10 cm layer, the mean $C_m$ and $C_0$ values were the highest in SS and the lowest in FS. Meanwhile, in the 10–20 cm layer, the highest $C_m$ and $C_0$ mean values were found in SS and MS, respectively. In contrast, the first-order rate constant of labile pool mineralization (k) was significantly lower in MS and SS compared to FS in the 0–10 cm layer (Table 2).

**Table 2.** Mean and standard error values of cumulative value of mineralized carbon during incubaT-able 35 days ($C_m$), potentially mineralizable carbon ($C_0$) and rate constant (k) results. The lowercase letters indicate differences between examined soil layers for individual land-use variants, while uppercase letters show differences between land-use variants separately for the 0–10 and 10–20 cm soil layers (Tukey post hoc test, $p < 0.05$). Standard error is reported behind "±".

| Layer (cm) | Land Use | $C_m$ (mgC-CO$_2$ g$^{-1}$) | $C_0$ (mgC-CO$_2$ g$^{-1}$) | k |
|---|---|---|---|---|
| 0–10 | Meadow | 20.58 ± 1.72 [Aa] | 25.73 ± 4.66 [Aa] | 0.081 ± 0.007 [Ab] |
|  | Succession | 22.17 ± 2.69 [Aa] | 32.75 ± 5.23 [Aa] | 0.059 ± 0.011 [Aa] |
|  | Forest | 18.14 ± 1.53 [Aa] | 18.93 ± 1.59 [Aa] | 0.095 ± 0.002 [Bb] |
| 10–20 | Meadow | 16.97 ± 2.55 [Aa] | 33.76 ± 6.11 [Aa] | 0.045 ± 0.012 [Aa] |
|  | Succession | 17.96 ± 2.62 [Aa] | 26.68 ± 4.55 [Aa] | 0.047 ± 0.009 [Aa] |
|  | Forest | 13.91 ± 2.40 [Aa] | 24.30 ± 6.22 [Aa] | 0.063 ± 0.013 [Aa] |

*3.3. Biochemical Indicators*

The calculated values of the biochemical indicators slightly depended on land use and soil layer (Table 3). The qCO$_2$ mean value was almost two times higher in SS compared to MS and FS in the 10–20 cm layer. Mean qMIC decreased from MS to SS to FS in both examined soil layers; significant differences were found between soils in the 10–20 cm layer. Mean qM slightly decreased from MS to FS in the 0–10 cm layer, whereas in the 10–20 cm layer the qM mean value in SS was about 47% higher compared to FS. Moreover, the mean values of qCO$_2$ and qM in the 0–10 cm layer of SS were significantly lower than the 10–20 cm layer (Table 3).

**Table 3.** Mean and standard error values of metabolic quotient (qCO$_2$), microbial quotient (qMIC) and mineralization quotient (qM) results. The lowercase letters indicate differences between examined soil layers for individual land-use variants, while uppercase letters show differences between land-use variants separately for the 0–10 and 10–20 cm soil layers (Tukey post hoc test, $p < 0.05$). Standard error is reported behind "$\pm$".

| Layer (cm) | Land Use | qCO$_2$ (mgC-CO$_2$ mgMBC$^{-1}$ 24 h$^{-1}$) | qMIC (mgMBC gC$_{org}$$^{-1}$) | qM (mgC-CO$_2$ mg C$_{org}$$^{-1}$) |
|---|---|---|---|---|
| | Meadow | 3.60 ± 0.68 [Aa] | 6.12 ± 0.83 [Aa] | 0.76 ± 0.19 [Aa] |
| 0–10 | Succession | 4.19 ± 0.61 [Aa] | 4.47 ± 0.52 [Aa] | 0.62 ± 0.08 [Aa] |
| | Forest | 3.38 ± 0.37 [Aa] | 4.21 ± 0.39 [Aa] | 0.49 ± 0.06 [Aa] |
| | Meadow | 6.51 ± 1.77 [Aa] | 7.14 ± 1.34 [Ba] | 1.15 ± 0.33 [Aa] |
| 10–20 | Succession | 12.87 ± 3.93 [Ab] | 3.82 ± 0.61 [Aa] | 1.32 ± 0.30 [Ab] |
| | Forest | 6.35 ± 1.83 [Aa] | 3.83 ± 0.47 [Aa] | 0.84 ± 0.23 [Aa] |

*3.4. Parameters Affecting Soil Organic Carbon Content*

The overall regression equation indicated that the soil organic carbon content, across all studied soils, depended on MBC, qMIC, DOC and qCO$_2$ values, according to the equation:

$$\text{Corg} = 28.6 + 0.22\text{MBC} - 4.83\text{qMIC} - 0.02\text{DOC} - 0.21\text{qCO}_2 \tag{1}$$

The obtained model explained 83% of the variance in the dependent variable, and the standard error of estimation was 7.5. Nevertheless, when we took into consideration the regression equations for individual land-use variants, we can state that different microbial parameters and biogeochemical indexes shaped C$_{org}$ content depending on land cover and soil layer. Comparing the regression models for meadow, succession forest and old-growth forest soil for both examined layers, it was noted that the models for different land-use variants clearly differed between MS, SS and FS for the 0–10 cm layer, while, in the 10–20 cm layer, C$_{org}$ MS and FS depended on the same parameters (Table 4). The regression models explained 77% to 99% of the variation in C$_{org}$ in the individual land-use variants and soil layers.

**Table 4.** Total organic carbon (C$_{org}$) dependent variable regression for individual examined land-use and soil layer variants.

| Land Use Layer (cm) | Meadow | | Succesion | | Forest | |
|---|---|---|---|---|---|---|
| | 0–10 | 10–20 | 0–10 | 10–20 | 0–10 | 10–20 |
| **Free word** | −5.10 | 2.59 | 39.22 | 12.01 | 67.24 | 45.98 |
| **qM** | −0.39 | −0.09 | - | −0.01 | −0.30 | 0.23 |
| **C$_m$** | 2.90 | - | - | - | 2.04 | - |
| **DHA** | 0.41 | - | −0.39 | - | 0.57 | - |
| **qMIC** | - | −0.52 | −9.14 | −3.24 | −5.60 | −9.54 |
| **MBC** | - | 0.14 | 0.35 | 0.30 | - | 0.19 |
| **DOC** | - | −0.10 | −0.11 | −0.03 | −0.06 | −0.05 |
| **INV** | - | 0.38 | 0.09 | 0.10 | −0.04 | - |
| **qCO$_2$** | - | 2.41 | - | 0.16 | −7.18 | −2.75 |
| **Corrected R$^2$** | 0.77 | 0.93 | 0.95 | 0.97 | 0.99 | 0.95 |
| **Estimation error** | 10.11 | 2.52 | 4.16 | 1.53 | 0.79 | 2.17 |

qM—mineralization quotient; C$_m$—cumulative value of mineralized carbon during incubation time (35 days); DHA—dehydrogenase activity; qMIC—microbial quotient; MBC—microbial biomass carbon; DOC—dissolved organic carbon; INV—invertase activity; qCO$_2$—metabolic quotient.

## 4. Discussion

*4.1. Impact of Natural Forest Succession on Soil Organic Carbon*

Land-use change is one of the main factors influencing the organic carbon cycle. Many studies, Bell et al., Liu et al., Luo et al., Mganga et al. [19,35–37], highlighted the importance

of land-use conversion on soil organic carbon dynamics. Moreover, such changes in land cover and land use can have a significant impact on global carbon pools and fluxes [38]. While, here, only a slight increase in $C_{org}$ during natural forest succession was noted, this was also confirmed in a previous study [39]. However, according to estimations, even a marginal (0.01% annually) increase in $C_{org}$ content in soil via carbon sequestration may easily offset annual rises in the atmospheric carbon dioxide [40]. Thus, even small changes may play an important role in carbon dynamics, carbon cycle and carbon sequestration at both the local and global scale.

Natural succession is the default management strategy for the abandonment of agricultural lands, which influence both aboveground and belowground carbon accumulation [41]. Moreover, natural forest succession increases forested areas, which are considered an important soil organic carbon pool. Forest soils contain almost half of the total organic carbon in terrestrial ecosystems [42] and play a crucial role in the context of carbon sequestration; therefore, increasing forest areas is recommended by current international policy agendas. On the other hand, overgrowth of highly valuable semi-natural meadows in Polish national parks in the Carpathian Mountains, caused by a continuous process of natural forest succession, can decrease biodiversity and cause the disappearance of unique mountain landscapes [43,44]. The differences in meadow and forest ecosystem stability compared to succession forests has been highlighted in previous studies [39,45]. However, the present study showed a greater similarity of $C_{org}$ changes in MS and FS in the 0–10 cm layer, while, in the 10–20 cm layer, the factors shaping $C_{org}$ in MS and SS were the same. Such results may confirm that the 0–10 cm soil layer is more susceptible to changes along land-use and land cover changes, such as natural forest succession.

*4.2. The Soil Properties and Biochemical Parameters Affecting Soil Organic Carbon Content across Changing Land Uses*

The influence of studied soil properties and calculated parameters on $C_{org}$ across all land uses in the Polish Carpathians was more differential in the 0–10 cm layer compared to the 10–20 cm soil layer. Forests and grasslands are assumed to be stable ecosystems, playing a crucial role in carbon sequestration and the carbon cycle. Moreover, grasslands, through photosynthesis, capture about 20% of the $CO_2$ released to the atmosphere annually [46]. Thus, the regression results of our study showed that the content of $C_{org}$ in FS and MS in the 0–10 cm layer is determined by some similar parameters. The positive impact of DHA in the regression model of $C_{org}$ in MS and FS highlights an important role of microbial activity in shaping the organic carbon pool in such ecosystems. Maini et al. [47] related that DHA levels vary in soils of different land uses with the addition of organic residues on the soil surface. However, Błońska et al. [48] found a large amount of component of nutrient cycling in the initial stages of organic matter decomposition in forest systems, which resulted in higher DHA levels. Meanwhile, in the 0–10 cm layer of SS, the INV level had significant influence on the $C_{org}$ content. Invertase is one of the most important enzymes in the soil carbon cycle and is related to the transformation and decomposition of soil organic carbon by hydrolyzing carbohydrates to sugars and oligomers that are suitable for uptake by plants and microbes [49].

The positive influence of INV and MBC on $C_{org}$, as well as the fact that the lowest mean values of these parameters were found in SS, may indicate that the presence and activity of microorganisms in such a constantly changing environment is the limiting factor for $C_{org}$ changes in succession land use. According to Souza et al. [50], the ability of microbial biomass to convert organic carbon under stress is reduced, resulting in decreased qMIC levels. Natural forest succession caused the deterioration of conditions for microbial growth, which was confirmed by the decrease in qMIC from MS to FS, as well as the negative impact of qMIC in the regression of $C_{org}$ in SS and FS. In contrast, Susyan et al. [51] noted an increase in qMIC levels within secondary forest succession. However, Insam and Domsch [52] reported declining qMIC in open-pit mine soils under forest succession, which indicated decreasing $C_{org}$ availability due to progressive accumulation of recalcitrant

humic material. Additionally, $C_{org}$ in the 0–10 cm layer for MS and FS was significantly influenced by mineralization parameters (qM and $C_m$), which may also confirm the higher stability of such ecosystems.

Bakhshandeh et al. [16] used qM as an indicator of chemical, biochemical and microbiological soil properties' stability. Meanwhile, here, natural forest succession influenced an increase in $C_m$ and $C_0$ content in the 0–10 cm layer, which indicated faster mineralization of organic matter in SS compared to FS and MS. According to Moscatelli et al. [33], lower values of $C_m$ and $C_0$ demonstrate lower respiration rates and the presence of a smaller fraction of organic carbon available for mineralization. Nevertheless, the rate constant of labile pool mineralization in SS was significantly lower compared to FS. Similarly, Jiang et al. [53] found the highest k values in forest land compared to other land-use types.

Conversely, in the 10–20 cm layer, the influence of natural forest succession on organic carbon changes was inconsiderable; especially in the MS and SS, the same parameters defined the $C_{org}$ content in regression models. However, the $qCO_2$, which is a frequently used indicator of ecosystem development and soil disturbance [54], was the most important parameter that influenced $C_{org}$ changes in Carpathian soils, especially in the 10–20 cm layer. SS had the highest $qCO_2$; similar results were presented by Susyan et al. [51], who found the highest $qCO_2$ values in succession stages on the arable lands. Susyan et al. [51] also stated that the lack of a decline in $qCO_2$ values during forest succession confirmed simultaneous changes in microbial biomass and respiration activity. The MS and FS in the Carpathians had a higher microbial efficiency in utilizing the available resources, which was related to the lower $qCO_2$ [19].

Additionally, lower $qCO_2$ in MS and FS may reflect the higher stability of organic substrates by the presence of microbial biomass [55]. However, the highest $qCO_2$ in SS was related to the lowest MBC, especially in the 10–20 cm layer. Such results are in agreement with Umarov et al. [56], who showed an increase in microbial biomass carbon content during the first 17 years after secondary forest succession, which then gradually decreased and remained at a quite constant level after 24 years of succession. Nevertheless, Susyan et al. [51] stated that microbial biomass increase could be related to increasing input and accumulation of organic carbon during forest succession, and, in disagreement with our results, they noted an increase in microbial biomass carbon along succession. Likewise, Mganga et al. [19] found lower qM values in the natural ecosystem in the Mt. Kilimanjaro region compared to arable land, which was associated with lower ratios of easily mineralizable organic matter to stable organic matter in soils under natural vegetation. These findings are in accord with our results obtained for FS, expressed by the qM values and regression models.

## 5. Conclusions

Despite no significant differences in the content of $C_{org}$ between semi-natural meadow, succession and old-growth forest, the natural forest succession in the Polish Carpathian Mountains influences the type and the rate of the $C_{org}$ decomposition and transformation. The obtained results have shown that the natural forest succession caused a decrease in MBC content, especially in the 10–20 cm layer. Moreover, SS characterized less efficient use of organic substrates by microbial biomass compared to MS and FS, expressed by $qCO_2$.

Overall, across all studied soils, the significant impact on $C_{org}$ had MBC, qMIC, $qCO_2$ and DOC. The obtained model explained 83% of the variance in $C_{org}$. Meanwhile, the content of $C_{org}$ in the individual land-use variants and depths was shaped by different biochemical factors depending on land use and soil layer. In the 0–10 cm soil layer, the differences between MS and SS, and FS and SS were more distinct, while the changes in $C_{org}$ in MS and FS were determined by similar properties. Meanwhile, in the 10–20 cm layer in MS and SS, the $C_{org}$ content was shaped by the same parameters.

The insight into $C_{org}$ changes confirmed the findings that meadows and forests are stable ecosystems that were formed over hundreds of years, while successional processes are dynamic continuously changing ecosystems—ecotone zones. The stability of meadow

and forest ecosystems is an important issue in the context of carbon sequestration and climate change. Taking into account the problems of decision makers managing protected areas, such as national parks, it would be appropriate to protect the semi-natural mountain meadows to preserve their biodiversity and the ecosystem services they provide, which are important globally, but even more at the local scale.

**Author Contributions:** Conceptualization, J.S., A.J. and T.Z.; methodology, J.S. and A.J.; software, not applicable; validation, J.S. and A.J.; formal analysis, J.S. and A.J.; investigation, J.S.; resources, J.S.; data curation, J.S.; writing—original draft preparation, J.S.; writing—review and editing, A.J. and T.Z.; visualization, J.S.; supervision, A.J. and T.Z.; funding acquisition, T.Z. All authors have read and agreed to the published version of the manuscript.

**Funding:** This research was funded by Statutory financial support of Ministry of Science and Higher Education RP Department of Soil Science and Agrophysics [010013-D011 in 2021] University of Agriculture in Krakow.

**Data Availability Statement:** The data presented in this study are available on request from the corresponding author.

**Acknowledgments:** We wish to thank Cambridge Proofreading team for reading the manuscript and for language corrections.

**Conflicts of Interest:** The authors declare no conflict of interest.

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
