# Peer review of "Impact of Natural Forest Succession on Changes in Soil Organic Carbon in the Polish Carpathian Mountains"

_forests, doi:10.3390/f13050744_

Round 1
Reviewer 1 Report
This manuscript is an interesting study to explore the impact of natural forest succession on soil organic carbon in the Carpathians of Poland. Although the manuscript tries to explain this interesting scientific problem, the current state of the manuscript is not suitable for publication. It is suggested to reconsider after major modifications. In addition, it is suggested to find a professional native English researcher to modify the language of the manuscript.
The details are as follows:
1) In the materials and methods, the whole is mixed together, which is very chaotic. It is suggested to list the research area, the methods used in this study, sampling scheme and experimental analysis, so as to grasp important information directly with readers and researchers.
2) In the results, it is suggested to summarize the results into several topics. At present, it is difficult for readers to directly capture the important information of the article.
3) The same problem exists in the discussion, as well as in the manuscript. Some expressions have problems. It is necessary to list the names of researchers, not just the numbers of references. This is very unprofessional. For example, 299 lines, 300 lines, 305 lines, etc.
Author Response
This manuscript is an interesting study to explore the impact of natural forest succession on soil organic carbon in the Carpathians of Poland. Although the manuscript tries to explain this interesting scientific problem, the current state of the manuscript is not suitable for publication. It is suggested to reconsider after major modifications. In addition, it is suggested to find a professional native English researcher to modify the language of the manuscript.
Thank you for your review and comments.
We would like to highlight, that the manuscript was proofread and edited by Cambridge Proofreading LLC.
The details are as follows:
1) In the materials and methods, the whole is mixed together, which is very chaotic. It is suggested to list the research area, the methods used in this study, sampling scheme and experimental analysis, so as to grasp important information directly with readers and researchers.
Done, we added the subheadings in the material and methods section and organized the text.
2) In the results, it is suggested to summarize the results into several topics. At present, it is difficult for readers to directly capture the important information of the article.
Done, we added the subheadings in the results section.
3) The same problem exists in the discussion, as well as in the manuscript. Some expressions have problems. It is necessary to list the names of researchers, not just the numbers of references. This is very unprofessional. For example, 299 lines, 300 lines, 305 lines, etc.
Done! We added the subheadings in the discussion section and ordered the text. We listed the names of researchers.

Reviewer 2 Report
It was a pleasure to read this interesting research. The natural forest succession is not only characterized by a decrease in microbial biomass carbon content but often by a decrease in species richness, also.
Thank you for this nice presentation of your results and wish you luck in the further review process.
Author Response
Comments and Suggestions for Authors
It was a pleasure to read this interesting research. The natural forest succession is not only characterized by a decrease in microbial biomass carbon content but often by a decrease in species richness, also.
Thank you for this nice presentation of your results and wish you luck in the further review process.
Thank you for your review!

Round 2
Reviewer 1 Report
The problems mentioned have been modified and it is suggested to publish the current version.